# An Overview on the Properties of Ximenia Oil Used as Cosmetic in Angola

**DOI:** 10.3390/biom10010018

**Published:** 2019-12-20

**Authors:** Gabriel Satoto, Ana Sofia Fernandes, Nuno Saraiva, Fernando Santos, Nuno Neng, José Manuel Nogueira, Tânia Santos de Almeida, Maria Eduarda Araujo

**Affiliations:** 1CQB, CQE, and Department of Chemistry and Biochemistry, Faculty of Sciences, University of Lisbon, Campo Grande, 1749-016 Lisbon, Portugal; gfsatoto@hotmail.com (G.S.); fjsantos@fc.ul.pt (F.S.); ndneng@ciencias.ulisboa.pt (N.N.); jmnogueira@ciencias.ulisboa.pt (J.M.N.); 2CBIOS—Universidade Lusófona’s Research Center for Biosciences and Health Technologies, Campo Grande 376, 1749-024 Lisbon, Portugal; ana.fernandes@ulusofona.pt (A.S.F.); nuno.saraiva@ulusofona.pt (N.S.); tania.almeida@ulusofona.pt (T.S.d.A.)

**Keywords:** ximenia oil, chemical composition, viscosity, UV transmission, cytotoxicity, keratinocytes

## Abstract

Ximenia (*Ximenia americana* L.) is a shrub, or small tree, native from Africa and spread across different continents. In Angola, the seeds oil is used by local populations, to prevent sunburn, to smooth and hydrate the skin, and to give it a pleasant color and elasticity, to prevent stretch marks, in pregnant woman, and also as hair conditioner. Herein, an oil sold in the region (LPO), and two others extracted in laboratory, from seeds collected in the same region, were investigated in terms of their composition, chemical properties, UV transmission. The three oils are similar although the LPO is more acidic, 0.48 mg KOH/g. GC-MS analysis indicated that the major components are the fatty acids, oleic (31.82%), nervonic (11.09%), ximenic (10.22%), and hexacosa-17,20,23-trienoic acids (14.59%). Long chain fatty acids, n ≥ 20, accounted for 51.1% of the total fatty acids. A thin film of the oil showed a reduction in transmittance from 200 to 300 nm. Viscosity studies of the LPO indicated that at normal temperature of skin, the oil can be spread over the skin as a thin film. At concentrations up to 10 µg/mL, the LPO is not toxic to human keratinocytes, suggesting the safety of this oil.

## 1. Introduction

Ximenia (*Ximenia americana* L.), known as wild plum, is a shrub, or small tree, belonging to the Olocaceae family. This shrub, native from tropical Africa, is nowadays spread across different continents and can be found in the Western Islands, Pacific Islands, New Zealand, and Central and South America. It is an evergreen tree providing flowers and fruits all year. It is drought resistant and grows in poor soils. The fruit has a large kernel that contains about 60% in oil [1]. Several studies have been published showing the medicinal potential of this plant, such as for the treatment of the skin [2], as antimicrobial [3], as gastroprotective action [4], and antioxidant activity [5]. 

In Angola ximenia is known as mupeke. Native populations use the wood as fuel, to produce long sticks appropriate to mix food while cooking and also to produce other domestic objects like spoons and handles for scythes because of its strength and attractive appearance. However, the most valuable part of this tree is the oil obtained from its seeds. In southwest Angola, the oil is extensively used by local populations, mainly as a skin cosmetic: to smooth and hydrate, to give a more pleasant color, to increase skin elasticity, to prevent stretch marks in pregnant woman, and as a hair conditioner [6]. The oil is also used as a folk medicine. Massage with the oil is claimed to cure arthritis and abdominal pains and to prevent the onset of varicose veins. There is some controversy about the edibility of the oil extracted from seeds. For instance, in Niger some authors considered it edible [7]. It was also reported that the oil is used in Angola to prepare food [8]. However, some authors [9] disagree with this use because of its physicochemical characteristics, namely the high peroxide content and the low saponification and iodine values. 

In southwest Angola, rural communities are dependent on cattle breeding for survival and to get some income to acquire industrial goods. In these societies, the possibility to get extra income collecting and selling forest products other than wood is very important. In this context, ximenia oil is traded in the local markets since it is appreciated as a skin cosmetic. It is also commercialized in urban markets, improving the economic sustainability of the populations that produce it [6].

The extraction of the oil by Angolan woman has recently been reported [6,8] and comprises several steps: drying the fruits, toasting the seeds, gridding the toasted seeds, and drying the resulting mass at the sun. Then this mass is toasted again and, after that, is cooked with hot water until the oil separates. The mixture is cooled to room temperature and the oil is collected by woman. The oil is again warmed during an established time and, after cooling, is placed in a container to be sold.

In this work three oil samples from ximenia, each obtained by different methods, were investigated: a locally produced oil (LPO), which was acquired in Bibala province of Namibe, in Southern Angola, and two others extracted in the laboratory from seeds collected in the same region, to compare the impact of different extraction procedures. Among the two oils extracted in the laboratory, one was produced using a procedure that mimics the traditional production and is herein designated as pseudo-artisanal oil (PAO). The other one was obtained by Soxhlet extraction and so is named Soxhlet extraction oil (SEO). The three oils were chemically characterized, the transmittance of UV radiation was investigated as well as their viscosity. To the best of our knowledge, this is the first time that this handcrafted ximenia oil (LPO), locally produced by Angolan population, using a hot procedure, is characterized and its properties compared with two other oils extracted in laboratory. Furthermore, since one of the major claims for the use of this LPO is as a skin cosmetic, the viscosity and cytotoxicity, in human keratinocytes (HaCat cells) was evaluated for the first time for this locally produced ximenia oil. 

## 2. Materials and Methods 

### 2.1. Chemicals

Dulbecco’s Modified Eagle’s Medium (DMEM) was obtained from Biowest (Nuaillé, France). Trypsin, penicillin-streptomycin solution, fetal bovine serum, and crystal violet (CV) were purchased from Sigma-Aldrich (Steinheim, Germany). Dimethylsulphoxide (DMSO) was purchased from Merck (Darmstadt, Germany). All other chemicals used were acquired from Sigma-Aldrich (Steinheim, Germany) and were Pro-analysis grade.

### 2.2. Oil Extraction

The fruits of *X. americana* were collected in municipality of Bibala, province of Namibe, in Southern Angola. Samples of the leaves were also collected, and photographs of the whole shrub were obtained. For botanical identification this material was compared with the herbarium specimens LISC024707 and LISC024683 belonging to Arquivo Científico Tropical Digital (ACTD) do Instituto de Investigação Científica Tropical (IICT), Lisbon.

#### 2.2.1. Local Extraction of the Oil (LPO)

Fruits were attained, in August 2016, from a local woman. This person was also responsible for obtaining the LPO used in this study, obtained through the traditional procedure previously described in the literature [8]. Briefly, the fruits were dried, the seeds were separated and part of it (2 kg) were used to locally obtain 330 mL of the oil. The yield of this process was 15%. The remaining part of the seeds were used in the laboratory to obtain the other oil samples.

#### 2.2.2. Simulation of the Artisanal Process (PAO)

This process of extraction intends to simulate the local artisanal procedure [8]. The seeds brought from Angola were placed in an oven at a temperature of 50 °C for 48 h before extraction, to facilitate the dehydration process. After that time, they were toasted in an aluminum pan. The toasted seeds were crushed mechanically until an oily paste was obtained. Then this paste was carefully toasted again. Next, distilled water was added and the mixture was transferred to another container and maintained at 100 °C with continuous stirring at 50 rpm for 1 h.

After that time the residual liquid was removed, cooled to room temperature, and filtered under vacuum. The filtrate was poured in a separating funnel to separate the two non-miscible liquids. Finally, the oil was separated from the aqueous layer, dried over anhydrous sodium sulfate and then placed in a glass bottle for further characterization. The process was repeated three times and the yield (mean ± STD) was 9.72 ± 0.19% (*v*/*w*).

#### 2.2.3. Solvent Extraction with a Soxhlet Apparatus (SEO)

In the laboratory, seeds were dried in an oven at 80 °C, overnight. Then 100 g were crushed, divided in three samples of the same weight (29.333 g) and each one was extracted with 100 mL of *n*-pentane during 18 h in a Soxhlet apparatus. The yield in oil (mean ± STD) was 18.123 ± 0.084 g which corresponds to 61.78 ± 0.53% (*v*/*w*).

### 2.3. Characterization of Ximenia Oil

#### 2.3.1. Determination of Density 

Density was determined by the pycnometer method using a 25 mL glass pycnometer. The LPO density values were determined at four different temperatures and they are the mean of three independent determinations. Values are presented in Table 1.

#### 2.3.2. Determination of Acid Value 

The acid value IA was determined according to the British Pharmacopeia [10]. Values are presented in Table 1.

#### 2.3.3. Percentage of Free Fatty Acid (FFA)

The percentage of FFA (as oleic acid) was obtained multiplying the acid value by the factor 0.503: %FFA = IA × 0.503 [11]. Values are presented in Table 1.

#### 2.3.4. Determination of Saponification Value

The saponification value Sv (mg KOH/g oil) was determined according to the A.O.A.C., 17th Ed, 2000, Official method 920.160, as is described in Manual of Methods of Analysis of Foods [12]. Values are presented in Table 1.

#### 2.3.5. Determination of Iodine Value

The iodine value (g I_2_/g oil) was determined as described by some authors [11]. Values are presented in Table 1.

#### 2.3.6. Determination of Peroxide Value

The peroxide value (mg O_2_/kg oil) was determined as described elsewhere [11]. Values are presented in Table 1.

#### 2.3.7. Determination of Viscosity

The viscosity of samples was measured with a TA Instruments AR1500ex rheometer using plate/plate geometry for selected temperatures relevant for the intended use. The temperature (30 °C, Table 1) is controlled by a Peltier system existing on the lower 100 mm diameter stainless steel plate with a 0.01 K resolution, the thermometer being located adjacent to the center of the plate just below the sample layer. The upper geometry used for all samples was a 40 mm diameter stainless steel plate.

The sample volume required for the selected gap (about 1000 µm) was loaded on the geometrical center of the lower plate and the upper plate was sent to the grim gap (selected gap + 200 µm). The final gap was adjusted to meet the proper filling of the geometry.

### 2.4. Evaluation of UV Absortion

UV spectra of the three oils was obtained by deposition of a thin layer (0.6 mg/cm^2^) over a quartz plate [13]. Spectra were acquired in a Shimadzu UV 1603 spectrophotometer with the cell compartment thermostatized at 35 °C.

### 2.5. Chemical Composition

The chemical composition of the oils was determined by GC-MS after derivatization of the triacylglycerides in the corresponding methyl esters and is presented in Table 2. 

#### 2.5.1. Saponification Followed by BF3-Catalyzed Methylation

A lipid sample in a 10 mL glass balloon (48.6 mg) was hydrolyzed with 3 mL of NaOH methanolic solution (0.5%) at 60 °C for 20 min. The reaction mixture was methylated with 5 mL of BF3/MeOH (14%) and heated until ebullition for 5 min. Twenty milliliter of saturated NaCl aqueous solution was added to the reaction product and extracted with 20 mL of *n*-hexane. The organic phase was dried with 1 g of magnesium sulfate (Merck, Darmstadt, Germany) and filtered using a 0.45 μm nylon syringe filter. The excess of *n*-hexane was evaporated until 0.5 mL under a gentle stream of purified nitrogen (>99.5%).

#### 2.5.2. GC-MS Analysis

GC-MS analyses were carried out on an Agilent 6890 Series gas chromatograph equipped with a 5973 N mass selective detector (USA). A programmed temperature vaporization injector operating in the split mode (1:100) at 250 °C and a ZB-5 capillary column (30 m × 0.25 mm i.d. 0.25 μm film thickness; 5% diphenyl, 95% dimethylpolysiloxane; Phenomenex, USA) were used. The increase in the oven temperature was programmed from 80 °C (held for 1 min) to 300 °C (held for 10 min), at a rate of 20 °C/min. Helium was used as carrier gas at constant pressure (9.8 psi) and the injection volume was 1 μL. The transfer line, ion source and quadrupole temperatures were maintained at 280, 230 and 150 °C, respectively, and a solvent delay of 4 min was selected. Electron ionization was performed at 70 eV and the mass spectrometer was operated in the full scan mode in the range 35–550 Da. All results were compared with Wiley’s library reference spectral bank (G1035B; RevD.02.00). Data recording and instrument control were performed by the MSD ChemStation software (G1701 CA; ver.C.00.00; Agilent Technologies, Little Falls, DE, USA.

### 2.6. Cell Culture and Cytotoxicity Assessment

HaCat human keratinocytes were routinely cultured in DMEM supplemented with 10% fetal bovine serum, 100 U/mL penicillin and 0.1 mg/mL streptomycin. Cells were kept at 37 °C, under an atmosphere containing 5% CO_2_ in air [14]. Approximately 5 × 10^3^ cells were cultured in 200 μL of culture medium per well, in 96-well plates, and incubated 24 h at 37 °C under a 5% CO_2_ atmosphere. Cultures were then treated with the extract (0.5–10 μg/mL) for a 24 h-period. The low water solubility of the oil precluded the evaluation of higher concentrations. The sample was initially solubilized in DMSO and then further diluted in PBS. The final concentration of DMSO in culture medium was 0.5%, except for the positive control (DMSO 5%). After the incubation, the crystal violet assay was carried out as previously described [15]. Two independent experiments were carried out, each comprising four replicate cultures.

## 3. Results and Discussion

In this work the physical and chemical properties of ximenia oil, obtained by three distinct procedures, were investigated to understand if the oil produced and sold in the local mark (LPO) is very different from the one extracted using an organic solvent (SEO). Additionally, when considering the use of this LPO as a cosmetic, other relevant properties were also evaluated, such as the cytotoxicity, acidity, and viscosity.

Considering the preparation of the oil samples, the first oil, named in this work as local production oil (LPO) was purchased in Bibala, to a local producer from the province of Namibe, in Southern Angola. This oil was produced using the traditional procedure used by women in that region. The second oil, named pseudo-artisanal oil (PAO) was produced in the laboratory using a procedure that intends to mimic the artisanal process, but by controlling some parameters such as the temperature of the oven and the drying time of the seeds. The third oil (SEO) was extracted with *n*-pentane using a Soxhlet extractor. PAO and SEO were obtained from the seeds of ximenia provided by the same woman that produced LPO. Regarding the yield of extraction of the three oils (Table 1), the best result was obtained with Soxhlet extraction of the raw seeds, SEO, with a yield of 62% (*v*/*w*). On the other hand, when considering the process that begins by toasting the seeds and then extracting the oil with hot water, it is interesting to point out that the procedure made by the local Angolan woman in the small villages, was slightly more efficient (15% yield), when compared to that obtained in the laboratory (10% yield). This fact is quite interesting, indicating that the local women were able to optimize their method of extraction using the technological conditions available to them.

### 3.1. Physical and Chemical Characterization

Characterization data of LPO, PAO, and SEO is presented in Table 1, alongside with literature data for the same parameters. Results show that there are no major differences between the three oils under study. In general, it can be said that SEO is a little more acidic since the acid value (0.59 mg KOH/g) is higher than the acid values of LPO and PAO, 0.48 and 0.51 mg KOH/g respectively. This value is similar to 0.56 mg KOH/g obtained recently by some authors [7]. However, all obtained values are higher than the value of 0.14 mg KOH/g reported by other authors [16], but much lower than the values of 3.4 mg KOH/g [17], obtained with the oil extracted from seeds collected in Kenya, and 16.13 mg KOH/g reported by Nigerian authors [18]. Although the oils described in previous references had been obtained by Soxhlet extraction with petroleum ether, all these different values are an indication that ximenia is very sensitive to environmental conditions, which may influence the acidity of the oil. Organoleptic properties, such as appearance, color, and odor are also similar among the three oils, with the LPO appearing slightly darker in color. Because of the small amount of seeds brought to the laboratory and the low amount obtained for PAO and SEO, the density was only obtained for LPO and is similar to the values described in the literature (Table 1).

### 3.2. Chemical Composition

The chemical composition of LPO, PAO, and SEO was obtained by GC-MS after derivatization to the corresponding methyl esters. Results are presented in Table 2.

Analysis of the fatty acid composition of the three oils, shows that they are all very similar, regardless of the process used to produce them. Twenty-two fatty acids were identified. The saturated fatty acids vary from 10.02 to 13.85%, for PAO and SEO, respectively. On the other hand, the % of monounsaturated fatty acids was 59.95% for PAO and 60.23% for LPO and SEO, while the polyunsaturated varied from 14.51% in SEO to 15.20% in PAO. The main component of the oils is oleic acid, which is present in 31.82 % for LPO, 33.99 % for PAO, and 36.15 for SEO. The other components present in large percentage are long chain fatty acids, namely: tetracosanoic acid, C_24_H_36_O_2_; ximenic acid, C_26_H_50_O_2_; and hexacosatrienoic acid; C_26_H_46_O_2_. According to our results, the percentage of fatty acids with a chain length ≥20 carbon atoms is about 50%. The oils also contain an acetylenic fatty acid, the ximenynic acid. These acids have already been described in ximenia seed oil [19,20]. For example, oleic acid accounted for 72% of the oil produced from seeds collected in Nigeria [19] while for the three oils studied in this work, this acid is present in 30–35%. This value is similar to the value of 32.5% found in an oil of commercial origin [21]. The existence of long chain fatty acids, namely lumepueic acid, C_30_H_58_O_2_, was also reported by the same authors [21], that found that this acid was present in 5.34%, while in this work this acid was present in a minor amount, 2.5% for LPO and 2.2% for PAO. Interestingly, for SEO, this long chain fatty acid was not detected, which may indicate that the solvent was not the most suitable for the extraction of this acid in particular or, although seeds were extracted for 18 h, the extraction time was still not enough. 

The acetylenic fatty acid ximenynic acid, which was first described in 1952 [22], was also present in the three oil samples, in about 6%. This acetylenic fatty acid is claimed to have anti-aging benefits. In fact, this acid is part of a commercial product tested on 19–34 years women, suffering from venous stasis and with main evidence of signs of cellulite on the thighs, and the results described by these authors showed that typical cellulitic skin parameters were evaluated with positive results [1,23,24]. 

Furthermore, previous studies have shown that oleic acid, the main component of ximenia oil, and palmitic acid, have high skin permeation and interact with subcutaneous lipids [1]. Behenic acid, C_22_H_44_O_2_, showed a positive effect on compromised skin, promoting a reduced transepidermal water loss and a more ordered lateral lipid packing in stratum corneum of patients with atopic dermatitis [25]. Finally, long-chain fatty acids in general are present in many patents that claim benefits in skin condition and anti-aging effects [26,27,28]. Also, since fatty acids have been described to have oiling, softening, smoothing, and protective properties, they have been classified to the group of emollients and have been successfully used in cosmetic products [29]. It is also interesting to acknowledge that the use of a procedure through heating (LPO and PAO) or using solvent extraction, at mild temperature conditions (SEO), does not seem to have a high impact on the chemical composition. 

### 3.3. UV Transmission Evaluation

Exposure to UV radiation has several negative effects on human skin such as, photo ageing, sunburn, and skin cancer. Solar UV radiation can be subdivided into three components, UVA comprising radiation from 400–320 nm, UVB comprising radiation from 320–280 nm, and UVC comprising radiation from 280–100 nm. Skin pigmentation is an adaptive process to overcome radiation damages and is usually assumed that dark skinned people are more protected against the harmful effects of UV radiation. Nevertheless, although black and other darker-pigmented populations are less prone to melanoma cancer than white populations, they are not completely protected against the UV radiation damages [30,31]. Thus, it is crucial to find ways to limit skin damage in consequence of UV radiation. Since one of the claims for the use of ximenia oil is its supposed protective effect against UV radiation, UV transmission spectra of the oils was performed in this study.

Results of the UV transmission spectra for the three oils (Figure 1), spread in a very thin film over a quartz plate, showed a drastic reduction in transmittance that goes from 200 nm to 300 nm. Furthermore, for the LPO sample, data also seem to show a slight decrease in transmittance from 200 nm up to 300 nm (UVC and beginning of the UVB region). Nonetheless, these results are not sufficient to ensure that this oil has UV radiation protective properties even though this ability is claimed by the Angolan population. 

Moreover, since the LPO is sold by local Angolan population as skin cosmetic, the viscosity of this oil, which may impact the consumers’ acceptance of the product, as well as its cytotoxicity, in human keratinocytes (HaCat cells), to assess its safety, were also evaluated.

### 3.4. Viscosity Studies

Rheological properties are important to predict the behavior and resistance of a fluid during the flow. Its determination is a useful tool in the process design and quality control in cosmetic and pharmaceutical industry. In fact, the viscosity of a product influences its acceptance by the consumer since the first impression is critical, because the consistency of the product is related to the perception of quality. Regarding vegetables oils, it is already known that viscosity increases with the increased content of very long-chain fatty acids and decreases with the increased content of polyunsaturated fatty acids [32]. Therefore, this rheological parameter is a function of the dimensions and orientation of the molecules present.

No pre-shear was used prior to measurements and the sample was allowed to properly adjust to the selected temperature (30 °C). A shear stress sweep was made to establish the flow behavior of the sample, from the lowest usable shear stress (0.008 Pa) to about 200 Pa, provision being made to prevent shear rates higher than 1000 1/s. If a Newtonian plateau is detected, then peak hold measurements may be made for some shear rates considered similar to values used by other type of viscometers.

The LPO sample presented a global shear-thinning, which means that the viscosity decreases with increasing shear rate. This behavior is common for most cosmetic products [32]. A Newtonian plateau was detected for shear rates between 5 and 1100 s^−1^, with viscosity of 0.132 ± 0.001 Pa s at 30 °C (roughly 165 times the viscosity of water) without any detectable signals of turbulent flow appearance.

Some Bingham type plasticity was detected with a yield stress of 0.015 Pa. The determination of the yield stress is also relevant, since the lower its value the easier the product can be spread on the skin, because the yield stress determines the thickness of the product layer on the skin. Additionally, the viscosity of the LPO decreases with temperature, as it is also normal for most cosmetic and personal care products [33]. 

### 3.5. Cell Culture and Cytotoxicity Assessment of LPO Oil

Since the native populations claim that the studied oil is useful as a cosmetic, it is fundamental to evaluate the cytotoxicity of the LPO in human normal-like keratinocytes (HaCat cells). The results of the cytotoxicity assay are depicted in Figure 2. 

Under our experimental conditions, the oil did not show cytotoxicity up to 10 μg/mL. Although this study suggests the safety of the studied oil, this claim is limited by the low concentration range tested, because of the low solubility of the oil. Nonetheless, our results are in line with the conclusions of the Cosmetic Ingredient Review Expert Panel which assessed the safety of fatty acid oils derived from 244 other plants and concluded they are safe as used in cosmetics [34].

## 4. Conclusions 

The ximenia oil produced by local Angola women is used by the population as a skin conditioner and as a UV screen. In this study three types of ximenia oil, an oil obtained from a local producer and two others extracted in laboratory, from seeds collected in the same region, were investigated, to assess if the local extraction could provide an oil with similar composition to the solvent extraction. 

The results attained from the chemical analysis, namely the acid, the saponification, the peroxide values, and the lack of toxicity to human keratinocytes, at the studied concentrations, are indicative that this oil may be suitable for topical use. Moreover, the claimed properties of some of the components found in the studied oil are indicative that it may be an interesting product to be used as a cosmetic base material in skin formulations. In fact, the oils showed a high content of oleic and ximenynic acids, that interact with subcutaneous lipids and have anti-aging effects, respectively. Also, the oils showed a high content in fatty acids that have protective properties and have been used in cosmetic products. Interestingly, despite the poor technological conditions, the LPO proved to have similar composition, compared to the oils obtained in laboratory, showing a quite efficient artisanal extraction method. The viscosity of the LPO oil decreased with temperature, similar to what happens with most cosmetic products and which is important for consumers acceptance. Hence, LPO shows potential to be used as a cosmetic base material for skin formulations.

## Figures and Tables

**Figure 1 biomolecules-10-00018-f001:**
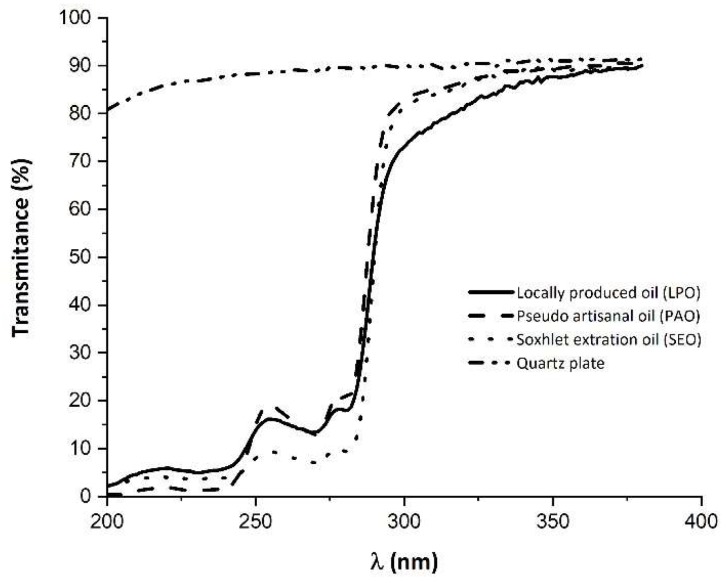
UV transmission spectra of the *Ximenia americana* oil obtained by the three different process. Transmission of the quartz plate is also displayed.

**Figure 2 biomolecules-10-00018-f002:**
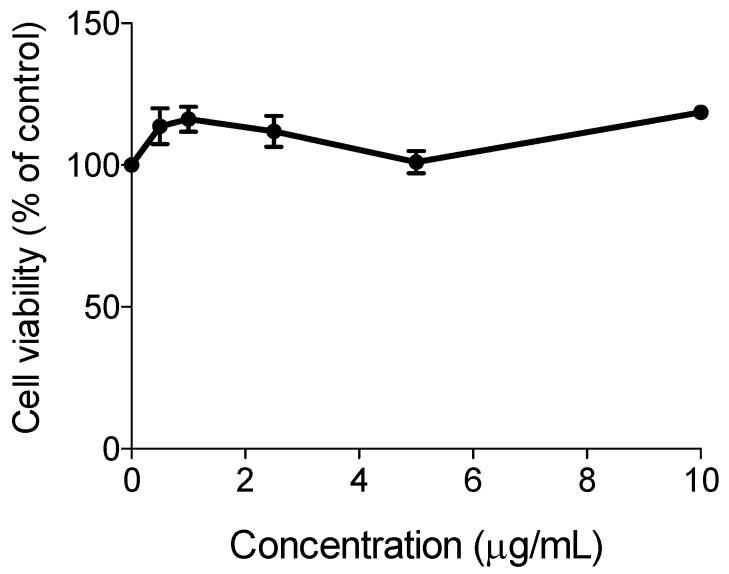
Effect of ximenia seed oil from artisanal production on the viability of human keratinocytes, as evaluated by crystal violet (CV) assay. Cells were incubated with increasing concentrations of the LPO oil for 24 h. Results are average values ± SD from two independent assays, each comprising four replicate cultures.

**Table 1 biomolecules-10-00018-t001:** Physico-chemical properties of the three ximenia oils obtained by different processes, LPO (local production oil), PAO (pseudo artisanal oil), and SEO (Soxhlet extracted oil).

	LPO	PAO	SEO	Comparison Literature
Yield (%, *v*/*w*)	15	9.72 ± 0.19	61.78 ± 0.53	45.7 [18] 54.5 [19]
Color	dark yellow	light yellow	light yellow	-
Odor	oily	oily	oily	-
Physical state at room temperature	liquid	liquid	liquid	-
Acid value (mg KOH/g)	0.48 ± 0.08	0.51 ± 0.07	0.59 ± 0.07	16.13 [18]; 3.4 [17]; 0.14 [16]; 0.56 ± 0.15 [7]
% of free fatty acid (FFA) (as oleic acid)	0.24 ± 0.01	0.26 ± 0.02	0.30 ± 0.01	8.07 [18]; 0.07 [16]; 0.29 ± 0.15 [7]
Mean molecular weight (g/mol)	757.55	741.05	746.67	-
Saponification value (mg KOH/g)	221.76 ± 0.10	229.67 ± 0.20	295.00 ± 0.17	178.5 [18]; 182.3 [16]; 179.94 ± 1.69 [7]
Iodine value (gI_2_/100g)	227.12 ± 0.09	281.51 ± 0.11	227.12 ± 0.10	152.28 [18]; 149.8 [16]; 40.61 ± 0.10 [7]
Peroxide value (mEq O_2_/kg)	30 ± 0.8	29 ± 1	31 ± 1.3	31.25 [18]; 29.4 [16]; 30.06 ± 0.12 [7]
Density (g/cm^3)^ T = 20 °C T = 30 °C T = 35 °C T = 40 °C	0.914 ± 0.0003 0.910 ± 0.0001 0.909 ± 0.0004 0.908 ± 0.0002	nd	nd	0.912 (30 °C) [18]; 0.974 (15 °C) [17]; 0.9493 ± 00 [7]
Viscosity (Pa.s, T = 30 °C)	0.132 ± 0.001	-	-	

nd—not determined.

**Table 2 biomolecules-10-00018-t002:** Chemical composition of the three types of ximenia oils obtained, LPO (local production oil), PAO (pseudo artisanal oil), and SEO (Soxhlet extracted oil).

Retention Time (min.)	FAME *	Chemical Formula of the Parent Acid	Oil		
	(Systematic Name/Common Name)		LPO	PAO	SEO
9.71	7-Hexadecenoic acid, methyl ester	C_16_H_30_O_2_	0.05	0.05	0.01
9.74	9-Hexadecenoic acid (palmitoleic acid), methyl ester	C_16_H_30_O_2_	0.06	0.02	0.01
9.83	Hexadecanoic acid (palmitic acid), methyl ester	C_16_H_32_O_2_	1.81	1.31	1.13
10.78	9-Octadecenoic acid (oleic acid), methyl ester	C_18_H_34_O_2_	31.82	33.99	36.15
10.89	Octadecanoic (stearic acid) acid, methyl ester	C_18_H_36_O_2_	0.99	0.89	0.80
10.93	9-Octadecynoic acid (stearolic acid), methyl ester	C_18_H_32_O_2_	0.26	Nd	Nd
10.95	9,12-Octadecadienoic acid (linoleic acid), methyl ester	C_18_H_32_O_2_	Nd	0.65	Nd
11.00	6,9-Octadecadienoic acid (isolinoleic acid), methyl ester	C_18_H_32_O_2_	0.04	0.04	Nd
11.19	7,10-Octadecadienoic acid, methyl ester	C_18_H_32_O_2_	0.08	Nd	Nd
11.38	(E)-octadec-11-en-9-ynoic acid (xymenicic acid), methyl ester	C_18_H_30_O_2_	5.89	6.66	5.17
11.75	11-Eicosenoic acid (gondoic acid), methyl ester	C_20_H_38_O_2_	2.09	1.81	1.59
11.84	Eicosanoic acid (arachidic acid), methyl ester	C_20_H_40_O_2_	0.33	0.38	0.28
11.92	11,14-Eicosadienoic acid, methyl ester	C_20_H_36_O_2_	Nd	1.73	Nd
12.68	13-Docosenoic acid (erucic acid), methyl ester	C_22_H_42_O_2_	2.42	2.44	1.97
12.78	Docosanoic acid (behenic acid), methyl ester	C_22_H_44_O_2_	1.18	1.18	1.15
13.76	15-tetracosenoic acid (nervonic acid), methyl ester	C_24_H_46_O_2_	11.09	10.15	10.64
13.87	Tetracosanoic acid (lignoceric acid), methyl ester	C_24_H_48_O_2_	2.98	2.63	3.52
15.15	(Z)-hexacos-17-enoic acid (ximenic acid), methyl ester	C_26_H_50_O_2_	10.22	9.32	9.87
15.28	Hexacosanoic acid (cerotic acid), methyl ester	C_26_H_52_O_2_	2.79	2.38	3.97
17.04	Hexacosa-17,20,23-trienoic acid, methyl ester	C_26_H_46_O_2_	14.59	12.79	14.51
17.21	Octacosanoic acid (montanic acid) methyl ester	C_28_H_56_O_2_	0.92	1.25	3.00
19.75	21-Triacontenoic acid (lumepueic acid), methyl ester	C_30_H_58_O_2_	2.50	2.18	Nd
% identified fatty acids			92.08	91.83	93.76
% saturated			10.99	10.02	13.85
% mono-insaturated			60.23	59.95	60.23
% poli-insaturated			14.71	15.20	14.51
% acetilenic			6.15	6.66	5.17
chain length (%)		18 ≤ n ≤ 16	40.99	43.59	43.26
		n ≥ 20	51.10	48.24	50.47

* FAME: fatty acid methyl esther; Nd: not detected.

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
