# Peer review of "An Overview on the Properties of Ximenia Oil Used as Cosmetic in Angola"

_biomolecules, 2019, doi:10.3390/biom10010018_

Round 1

Reviewer 1 Report

1.Many other papers about Ximenia have been reported. The other literatures or studies of Ximenia should be stated and cited in the “Introduction”. And the author should tell the novelty or any improvement of analytical aspects of this study.

2.The discussion of results of “UV transmission evaluation” should be rechecked. The major reduction in transmittance was between 200-280 nm (UVC range, which is absorbed by the ozone layer). However, it did not show good absorption in the UVA and UVB range; it should not be concluded as a good UV-filters in cosmetics.

3.The author should conclude which extractive method is best for Ximenia to be as a good material in cosmetics.

Author Response

COMMENTS TO REVIEWER #1:

Thank you so much for your valuable suggestions and guidance on our manuscript.

We, the authors, want to express our deep appreciation and we have revised our manuscript seriously and carefully.

The revised parts are listed below and highlighted in the main document.

Comment_1

“Many other papers about Ximenia have been reported. The other literatures or studies of Ximenia should be stated and cited in the “Introduction”. And the author should tell the novelty or any improvement of analytical aspects of this study.”

Answer_1

We agree with the suggestion of the Reviewer #1 and have included four new references in the Introduction of the manuscript. Furthermore, we have made it clearer the novelty of this manuscript at the end of the introduction and in the conclusion sections.

Comment_2

“The discussion of results of “UV transmission evaluation” should be rechecked. The major reduction in transmittance was between 200-280 nm (UVC range, which is absorbed by the ozone layer). However, it did not show good absorption in the UVA and UVB range; it should not be concluded as a good UV-filters in cosmetics.”

Answer_2

We have revised this section accordingly to the reviewer suggestions and made the proper modifications at the end of section “3.4. UV transmission evaluation”.

Comment_3

“The author should conclude which extractive method is best for Ximenia to be as a good material in cosmetics.”

Answer_3

We have revised the conclusion section accordingly to the reviewer suggestion.

Reviewer 2 Report

Line 81: Materials- it is common to list the city and state/country of each company/manufacturer in this section in brackets.

Line 126: results should not be discussed here

Line 129: the sentence referring to values should be removed from this section

Line 133: where did 0.503 come from?, the sentence referring to values should be removed from this section

Line 142: the sentence referring to values should be removed from this section

Line 146: the sentence referring to values should be removed from this section

Line 150: temperature at the time of measurements should be mentioned, it is critical to know this information

Line 160: where did the 2 mg come from? Which guidelines were the authors following?

Line 177: city, state/country should be included

Table 1: nd should be defined under the table

Line 296: what skin parameters are the authors referring to? This was not mentioned under methods.

Line 320: UVC does not reach the surface of Earth. It is a lethal radiation and is fortunately filtered out by the ozone layer. The authors should not highlight UVC protection in this paragraph because it does not make sense.

Figure 1. The image is blurry. A higher quality image should be used for even for a review.

Line 343: what was that temperature?

Line 353: what is “de” in the sentence?

Line 360: should be aligned to the left

Additional comment: the degree signs are not the proper signs for degree C in the manuscript, please use the proper symbol for this purpose

Author Response

COMMENTS TO REVIEWER #2:

Dear Reviewer, thank you so much for your valuable suggestions and guidance on our manuscript.

We, the authors, want to express our deep appreciation and we have revised our manuscript seriously and carefully taking your suggestions into account.

Comment_1

Line 81: Materials- it is common to list the city and state/country of each company/manufacturer in this section in brackets.

Answer_1

We added the requested information as suggested.

Comment_2

Line 126: results should not be discussed here

Answer_2

We agree and have performed the requested change.

Comment_3

The reviewer referred that values should be removed from the material section.

Answer_3

We agree and have performed the requested changes.

Comment_4

Line 133: where did 0.503 come from?, the sentence referring to values should be removed from this section

Answer_4

The acidity of an oil, both acid value (IA) expressed as mgKOH/ g of oil, and % of free fatty acids (FFA) can be evaluated by titration. The same experiment allows to obtain both values. By definition IA= 56.1VN/M and the % of FFA expressed in oleic acid is FFA(%)=28.2VN/M, where V is volume in mL of the titrant used, N is the normality of the KOH solution and M is the mass of the oil sample in g. When the two formulas are combined a math relation is obtained: IA/56.1= FFA(%)/28.2. The quotient between 28.2 and 56.1 is 0.503

Comment_5

Line 150: temperature at the time of measurements should be mentioned, it is critical to know this information

Answer_5

We agree and have performed the requested change.

Comment_6

Line 160: where did the 2 mg come from? Which guidelines were the authors following?

Answer_6

A reference was missing. We added the missing reference and present the procedure as in the literature cited.

Comment_7

Line 177: city, state/country should be included

Answer_7

We agree and have performed the requested change.

Comment_8

Table 1: nd should be defined under the table

Answer_8

We have included the requested information.

Comment_9

Line 296: what skin parameters are the authors referring to? This was not mentioned under methods.

Answer_9

We recognize that our sentence was confused, and we performed the proper correction.

Comment_10

Line 320: UVC does not reach the surface of Earth. It is a lethal radiation and is fortunately filtered out by the ozone layer. The authors should not highlight UVC protection in this paragraph because it does not make sense.

Answer_10

We agree and have performed the requested change.

Comment_11

Figure 1. The image is blurry. A higher quality image should be used for even for a review.

Answer_11

We have increased the quality of Figure 1.

Comment_12

Line 343: what was that temperature?

Answer_12

We have included this information

Comment_13

Line 353: what is “de” in the sentence?

Answer_13

It was a typo. We have corrected this.

Comment_14

Line 360: should be aligned to the left

Answer_14

We have performed as suggested.

Comment_15

Additional comment: the degree signs are not the proper signs for degree C in the manuscript, please use the proper symbol for this purpose

Answer_15

We have corrected it.

COMMENTS TO REVIEWER #2:

Dear Reviewer, thank you so much for your valuable suggestions and guidance on our manuscript.

We, the authors, want to express our deep appreciation and we have revised our manuscript seriously and carefully taking your suggestions into account.

Comment_1

Line 81: Materials- it is common to list the city and state/country of each company/manufacturer in this section in brackets.

Answer_1

We added the requested information as suggested.

Comment_2

Line 126: results should not be discussed here

Answer_2

We agree and have performed the requested change.

Comment_3

The reviewer referred that values should be removed from the material section.

Answer_3

We agree and have performed the requested changes.

Comment_4

Line 133: where did 0.503 come from?, the sentence referring to values should be removed from this section

Answer_4

The acidity of an oil, both acid value (IA) expressed as mgKOH/ g of oil, and % of free fatty acids (FFA) can be evaluated by titration. The same experiment allows to obtain both values. By definition IA= 56.1VN/M and the % of FFA expressed in oleic acid is FFA(%)=28.2VN/M, where V is volume in mL of the titrant used, N is the normality of the KOH solution and M is the mass of the oil sample in g. When the two formulas are combined a math relation is obtained: IA/56.1= FFA(%)/28.2. The quotient between 28.2 and 56.1 is 0.503

Comment_5

Line 150: temperature at the time of measurements should be mentioned, it is critical to know this information

Answer_5

We agree and have performed the requested change.

Comment_6

Line 160: where did the 2 mg come from? Which guidelines were the authors following?

Answer_6

A reference was missing. We added the missing reference and present the procedure as in the literature cited.

Comment_7

Line 177: city, state/country should be included

Answer_7

We agree and have performed the requested change.

Comment_8

Table 1: nd should be defined under the table

Answer_8

We have included the requested information.

Comment_9

Line 296: what skin parameters are the authors referring to? This was not mentioned under methods.

Answer_9

We recognize that our sentence was confused, and we performed the proper correction.

Comment_10

Line 320: UVC does not reach the surface of Earth. It is a lethal radiation and is fortunately filtered out by the ozone layer. The authors should not highlight UVC protection in this paragraph because it does not make sense.

Answer_10

We agree and have performed the requested change.

Comment_11

Figure 1. The image is blurry. A higher quality image should be used for even for a review.

Answer_11

We have increased the quality of Figure 1.

Comment_12

Line 343: what was that temperature?

Answer_12

We have included this information

Comment_13

Line 353: what is “de” in the sentence?

Answer_13

It was a typo. We have corrected this.

Comment_14

Line 360: should be aligned to the left

Answer_14

We have performed as suggested.

Comment_15

Additional comment: the degree signs are not the proper signs for degree C in the manuscript, please use the proper symbol for this purpose

Answer_15

We have corrected it.

Round 2

Reviewer 1 Report

The authors seem to consider all the reviewers’ comments and suggestions, and supplements have been made in the revised manuscript.